# Artificial Visual System for Orientation Detection Based on Hubel–Wiesel Model

**DOI:** 10.3390/brainsci12040470

**Published:** 2022-04-01

**Authors:** Bin Li, Yuki Todo, Zheng Tang

**Affiliations:** 1Division of Electrical Engineering and Computer Science, Kanazawa University, Kanazawa 9201192, Japan; crislee@stu.kanazawa-u.ac.jp; 2Faculty of Electrical, Information and Communication Engineering, Kanazawa University, Kanazawa 9201192, Japan; 3Department of Intelligence Information Systems, University of Toyama, Toyama 9308555, Japan; ztang@eng.u-toyama.ac.jp

**Keywords:** hubel–wiesel model, orientation selectivity, artificial visual system

## Abstract

The Hubel–Wiesel (HW) model is a classical neurobiological model for explaining the orientation selectivity of cortical cells. However, the HW model still has not been fully proved physiologically, and there are few concise but efficient systems to quantify and simulate the HW model and can be used for object orientation detection applications. To realize a straightforward and efficient quantitive method and validate the HW model’s reasonability and practicality, we use McCulloch-Pitts (MP) neuron model to simulate simple cells and complex cells and implement an artificial visual system (AVS) for two-dimensional object orientation detection. First, we realize four types of simple cells that are only responsible for detecting a specific orientation angle locally. Complex cells are realized with the sum function. Every local orientation information of an object is collected by simple cells and subsequently converged to the corresponding same type complex cells for computing global activation degree. Finally, the global orientation is obtained according to the activation degree of each type of complex cell. Based on this scheme, an AVS for global orientation detection is constructed. We conducted computer simulations to prove the feasibility and effectiveness of our scheme and the AVS. Computer simulations show that the mechanism-based AVS can make accurate orientation discrimination and shows striking biological similarities with the natural visual system, which indirectly proves the rationality of the Hubel–Wiesel model. Furthermore, compared with traditional CNN, we find that our AVS beats CNN on orientation detection tasks in identification accuracy, noise resistance, computation and learning cost, hardware implementation, and reasonability.

## 1. Introduction

The human brain nervous system is a highly complex deep network constructed by more than $10^{11}$ neurons [1]. About 80% of information received by our brain comes from the visual system, and the neurons in the human brain are more concentrated on visual tasks [2,3]. Accordingly, starting with research on the visual system is widely considered a proper way to figure out how the brain works. Phenomenally, forms, colors, and movements are fundamental and distinct attributes of visual images. Thus, we think the essential functions in our visual system include form perception, color perception, and motion perception [4,5]. In the visual system, we consider orientation detection a form perception. It has an essential role in human behavioral decision making. Whereas so far, the principle of orientation selectivity remains unclear [6,7]. Once the mechanism of visual orientation detection is understood, it would be of significance on the studies of the human brain [8,9]. From 1955 to 1978, Hubel and Wiesel systematically studied the visual functional structure [10]. In 1959, they reported that some cat cortical neurons showed orientation selectivity [11]. When they showed objects with various shapes and locations in front of a cat’s eyes, these neurons had the optimal response to an object with a specific orientation and specific location; otherwise, with little or no response [12]. In 1968, they reported that some neurons also had similar characteristics in the monkey’s striate cortex, but its optimal response to an object was no longer limited to a specific location. Hubel and Wiesel named these two kinds of neurons: simple cells and complex cells [13]. Simple cells are a simple type of cortical cells in the visual cortex, which have the optimal response to an object with a specific orientation at a certain fixed location in the visual field. Furthermore, an ineffective response area for a simple cell might be effective for another cell. Complex cells also have optimal responses to objects with a specific orientation angle but no longer limit objects’ locations in the vision field. The objects with optimal orientation can move in the receptive field without causing neuron inactivation [14,15,16,17]. To explain the orientation selectivity of these cortical cells, Hubel and Wiesel put forward a scheme that a simple cell’s receptive field can be integrated from the center-surround receptive fields of several LGN cells, and a complex cell’s receptive field is integrated from several simple cells’ [10,11]. Thus the neural circuit is constructed into a feedforward neural network which we usually call it Hubel–Wiesel (HW) model (described in Section 2.1). Although the HW model has not been fully proved, some physiological experimental results showed that indeed there were connections between LGN cells and simple cells, which proves some possibility of the HW model [18,19,20]. Simultaneously, with the increase in practical application requirements, there are several ways to realize orientation detection: principal component analysis method, gradient modeling method, digital filter method, and CNN method [21,22,23,24]. Among these methods, after training with a considerable number of data, the CNN method showed better recognizing performance [25,26]. With increasing requirements for more complex scenes, the traditional deep learning method falls into the generalization difficulty, and a trained model usually could be applicable for limited tasks. Thus, scientists have started to concentrate more on the brain and try to apply the visual information processing mechanisms to computer vision or artificial intelligence [27]. Although there are already many works focused on the simulation of the visual system, most of them could not directly be applied in deep learning. Some methods focus on the realization in an electronic device manner [28,29,30], though these electronic device-based works are impressive, they are hard to connect with computer vision. Some related research concentrated on the simulation of biology features [31,32,33], and though these systems can simulate the cell features, they were designed complicated. Some works concentrated on the application [34,35,36], and although they are applicable, without generalization, they are task-limited. So far, we lack a concise and efficient quantitative manner for the classical HW model.

To prove the reasonability and practicality of the HW model and explain the orientation selectivity in a qualitative manner, we propose a McCulloch-Pitts (MP) neuron-based orientation detective scheme and implement an artificial visual system (AVS) for two-dimensional object orientation detection. Simple cells and complex cells are realized by the MP neuron model. For simplicity, we realize four types of simple cells for a 3 × 3 two-dimensional local receptive field, each of which corresponds to a specific orientation angle ($0^{\circ },45^{\circ },90^{\circ }$, and $135^{\circ }$). In the detection process, the orientation information of each local receptive field is extracted by the simple cells separately and converged to the same type of complex cells. The function of the complex cell is to converge the activations of all simple cells. Finally, the global orientation is inferred by the activation degree of the complex cell. The type of complex cell with the most activation corresponds to the global orientation. Based on this scheme, we implement an AVS for global orientation detection, and its performance is evaluated by computer simulation on an image dataset. The objects in this dataset are two-dimensional and with different ideal shapes, locations, and orientation angles. The computer simulation results show that the AVS has biological similarities with the biological visual system and offers excellent orientation recognition accuracy to objects with different sizes, shapes, and locations, thus directly proving the reasonability and practicality of the HW model. To show the AVS’s superiority, we compare AVS and CNN’s performance on orientation detection and find that AVS beats CNN in identification accuracy, noise resistance, computation and learning cost, hardware implementation, and reasonability.

## 2. Mechanism and System

This section introduces the Hubel–Wiesel (HW) model and realizes simple cells and complex cells by artificial neuron model. Finally, we describe the implementation of an artificial visual system based on the HW model for two-dimensional object orientation detection.

### 2.1. Hubel–Wiesel Model

Hubel–Wiesel (HW) model is a scheme for explaining the orientation selectivity of cortical cells [10]. Orientation selectivity is a feature of cortex cells observed from Hubel and Wiesel’s experiments. The setup and results of the experiment on a cat are roughly shown in Figure 1 [17,37,38]. The electric signals show that some cells in the cat cortex were found to have the optimal response to which light stimuli with a specific orientational edge and a fixed location. These cells are named simple cells. Furthermore, another type of cells called complex cells respond vigorously to stimuli with a specific orientational edge but are no longer limited to a fixed local location, and the optimal orientational stimuli can make the complex cell activated the most within every location of the global receptive field [14,15,16,17].

To explain the orientation selectivity of simple cells and complex cells, Hubel and Wiesel proposed a feedforward model scheme. They speculated that simple cells receive the convergent input of several LGN cells whose receptive fields are arranged with a definite orientation. Thus the simple cell’s optimal response is tuned to stimuli with this specific orientation. Similarly, a complex cell’s inputs converged from several simple cells with the same orientation selectivity. Accordingly, complex cells can realize the insensitivity of stimuli location within global receptive field [10,11]. Figure 2 shows the process of receptive fields’ linking [17,38]. Thus, a classical HW feedforward model can be described theoretically as Figure 3.

### 2.2. McCulloch-Pitts Neuron Model

In the 1940s, McCulloch and Pitts proposed s simple model of biological neurons [39]. McCulloch-Pitts (MP) neuron model is a simplification of the biological nerve cells, which has only two states 1: excited (fire) and 0: not excited (inhibited). Figure 4 describes the detailed structure of an MP neuron. The neuron accepts inputs with different weights. When the weighted sum exceeds a certain threshold, the neuron will fire and output y = 1; otherwise, y = 0 [40].

### 2.3. Realization of Simple Cell and Complex Cell

This section describes the realization of simple and complex cells based on the artificial neuron model. Due to studies on biological neural networks, now we have the consensus that a single neuron can perform a simple task. Thus, we design simple cells based on the MP model for local orientation detection and complex cells with a sum function for summary activation according to the feature of simple cells and complex cells.

As we introduced above, a simple cell’s receptive field may be formed by several LGN cells’ receptive fields (see Figure 2a). In this paper, for the simplicity of the simulation implementation and neuron computation, we design each simple cell with a 3×3 local receptive field. As shown in Figure 5, several LGN cells’ receptive fields construct a simple cell receptive field size of 3×3. And for a simple cell, its activation directly depends on the light information in its spatial receptive field. Thus we decide to omit the processing of retinal cells in this pathway and let the light information directly transmitted into simple cells, so we do not need to consider which and how many LGN cells’ neural signals are inputted to a simple cell. When light falls on a region, a bunch of photoreceptors accepts the light signal and generates a corresponding electrical signal. Then the electrical signal is transmitted to simple cells through the primary visual pathway. The simplified signal transmission circuit is shown in Figure 6. We simplify using one photoreceptor to accept light information in a one-pixel region. Then the light information in the nine pixels (3×3) region is directly transmitted to a simple cell. The electrical signals are simplified to 0–1 signals. When a photoreceptor accepts light, it outputs 1; otherwise, 0.

This study introduces four kinds of orientation-selective simple cells based on the MP model for detecting orientation angles of $0^{\circ }$, $45^{\circ }$, $90^{\circ }$, and $135^{\circ }$, respectively. The details of a $45^{\circ }$-selective simple cell are as illustrated in Figure 7. In Figure 7, the input signal is expressed by $x_{i,j}$, where the ‘*i*’ and ‘*j*’ represent the two-dimensional location in the local receptive field. Furthermore, for the simplicity of the realization of AVS (introduced in next section) and neural computation, we idealize the ‘OFF’ regions of simple cells, light stimulation in the ‘OFF’ region will not cause any inhibitory response of the corresponding simple cell (the weights of neural connections in OFF region can be regarded as 0).

As shown in Figure 7, only the light stimulation in optimal orientation-selective region (ON region) is received by the $45^{\circ }$-selective simple cell. For a $45^{\circ }$-selective simple cell, the optimal orientation-selective locations are $x_{i,j}$, $x_{i+1,j-1}$ and $x_{i-1,j+1}$. The spatial light information in the local receptive field is projected on the retina and received by corresponding photoreceptors, generated electrical input signals are transmitted into the $45^{\circ }$-selective simple cell. When a photoreceptor receives light, the generated input is 1 (effective input), and its weight of neural connection to a simple cell is set to 1. Threshold θ is set to 2.5. When the weighted sum of inputs reaches the threshold θ, the neuron is activated. Thus, if and only if the $x_{i,j}$, $x_{i+1,j-1}$, and $x_{i-1,j+1}$ are all effective inputs, the $45^{\circ }$ simple cell is activated. The activation results can be expressed by the following equation:(1)$$y=\left\{\begin{array}{c} 1,\;(x_{i-1,j+1}+x_{i,j}+x_{i+1,j-1})\geq 2.5;\\ 0,\;(x_{i-1,j+1}+x_{i,j}+x_{i+1,j-1})<2.5. \end{array}\right.$$

The structures of the four types of orientation-selective simple cells and their optimal stimuli orientation are shown in Figure 8. Likewise, the simple cells in the other three orientations are realized in the similar way. In a 3×3 local receptive field, they also only respond to three effective inputs. $0^{\circ }$-selective simple cell only responds to stimulus in $x_{i,j}$, $x_{i,j-1}$ and $x_{i,j+1}$. The inputs of $90^{\circ }$-selective simple cell come from $x_{i,j}$, $x_{i-1,j}$ and $x_{i+1,j}$. $135^{\circ }$-selective simple cell’s effective input locations are set to $x_{i,j}$, $x_{i-1,j-1}$ and $x_{i+1,j+1}$.

Complex cells in our proposed detective scheme are responsible for converging the total activation of all simple cells. For simplicity, the simple cells with the same optimal orientation selectivity are connected to one single complex cell. Correspondingly, four different orientation-selective complex cells are needed ($0^{\circ }$, $45^{\circ }$, $90^{\circ }$, and $135^{\circ }$). The realization of a complex cell is described in Figure 9. The following equation can express the output result by the complex cell:(2)$$z=\sum_{i=1}^{n}y_{i}$$

### 2.4. AVS for Global Orientation Detection

We implement an artificial visual system (AVS) for two-dimensional object orientation detection based on the simple and complex cells we design. The AVS’s structure and the process of global orientation detection on an object by AVS are described in this section. As mentioned above, the simple cells can be activated by a 3-pixel optimal orientated line within a 3×3 local receptive field. For a large image size of M×N, take each pixel as a central point to divide this image into M×N local receptive fields. So the basic detection scheme for a large size image by AVS uses the simple cell to detect possible orientations of every local receptive field and uses complex cells to record the total activations of each type of simple cell. Accordingly, to extract local orientation information of an object in a two-dimensional M×N image, M×N×4 simple cells and 4 corresponding complex cells are needed.

Figure 10 shows the entire structure of AVS for detecting the global orientation of an object in a 5×5 image. For a 5×5 image, taking each pixel as the central location, it can be divided into 25 local receptive fields size of 3 (regions of the local receptive field beyond image can be regarded as no light stimulation in this region). The light stimulation in each local receptive field will accept by 9 photoreceptors and generate corresponding 0–1 signal inputs. In each local receptive field, the photoreceptors are connected with a set of four different simple cells. Each group of simple cells separately extracts the 25 local orientation information. Subsequently, activation results of all simple cells are input to corresponding same type complex cells. According to the function of complex cells we design, the complex cell can sum up the total activation of each type of simple cell to get final outputs representing the activation of four orientations. The object’s global orientation is inferred from the type of complex cell that is most activated. In Figure 10, we omit the presentation of simple cells for detecting the edge region (no simple cell be activated in the edge region). We can find that the activation result of $135^{\circ }$-selective complex cell is 5, and $45^{\circ }$-selective complex cell was not activated. $0^{\circ }$-selective and $90^{\circ }$-selective complex cell’s output value are 3. Thus, the final orientation of the object is determined by the complex cell ($135^{\circ }$-selective) with the most activations (5).

## 3. Simulation and Result

This section describes the validation results of AVS on datasets and some biology-inspired experiments. We also compared AVS and CNNs’ performance on noise data. All simulations were implemented on the Apple M1 chips hardware environment.

To validate the mechanism’s feasibility and the mechanism-based AVS, we implemented this mechanism and the AVS for global orientation detection by computer simulation. This section describes the AVS’s physiological similarity with the biological visual system and evaluates the practicality combined with dataset testing results.

We first tested the AVS’s feasibility on a binary image dataset. The images were sized to have 1024 pixels (32×32) and at least 3 pixel light spots. In each image, light spots were formed into an ideal object (central symmetry or axial symmetry) with a specific orientation angle ($0^{\circ },45^{\circ },90^{\circ }$, or $135^{\circ }$). We evaluated the detection system by analyzing its recognizing accuracy on 45,788 images, and the results are summarized in Table 1. The AVS has high detection accuracy for the orientation of ideal objects in binary images.

An example of the orientation detection on an object in a binary image is provided in Figure 11. From Figure 11a, we can observe that the object’s orientation angle in the image was $135^{\circ }$. This 32×32 image could be divided into 1024 local receptive fields. To get this object’s global orientation, 4096 (32×32×4) simple cells were needed to detect each local receptive field, and 4 complex cells were required to converge the activation of simple cells. Referring to the biologists’ potential recording method of a single neuron [11], we also used spike rate to record the complex cells’ activation degree. One active simple cell could let the same type of complex cell generate a spike, so theoretically, a complex cell’s spike rate could be up to 1024. From the activation results shown in Figure 11b,c, we can see that the $135^{\circ }$-selective complex cell’s spike rate was 44, which was the most. Thus, the orientation detection result of the object was $135^{\circ }$.

An example of detecting an $0^{\circ }$-object is shown in Figure 12. From Figure 12b,c, we know that the $0^{\circ }$-selective complex cell’s spike rate was 65, which was the most. The detection result was the same as we observed by our eyes. When we rotated the object in Figure 12a to other orientation angles and detected the object by our AVS, we could obtain the result shown in Figure 13. When an object was oriented at different orientations, it would activate the corresponding complex cell the most. This result also supports that our mechanism is feasible and reliable.

To further verify our mechanism, we conducted some comparative experiments. First, let us look at the object shown in Figure 14a. It is a 5 × 5 square. From the detection results shown in Figure 14b,c, we know that $0^{\circ }$-selective complex cell and $90^{\circ }$-selective complex cell spike rate were the same and were the most. So the AVS cannot determine which angle is the object’s orientation. It is the same for the biological visual system because we humans cannot tell its orientation angle. We can say it is oriented toward $0^{\circ }$ or $90^{\circ }$ at the same time. To further investigate the correlation between object shape and complex cell activation, we gradually increased the length of a 3 × 3 square along the direction of $0^{\circ }$ until it became a 3 × 18 rectangle. The objects and the spike rate curves are shown in Figure 15. Then we started to increase the length of the 3 × 18 rectangle along the direction of $90^{\circ }$ until it became a square again. The objects and the spike rate curves are shown in Figure 15. From the spike rate curve shown in Figure 15b, we can observe that as the object’s shape became more and more inclined to a rectangle with distinguishable length and width, the spike rate between complex cells became markedly different. The spike rate of the $0^{\circ }$-selective complex cell far exceeded other cells, which means AVS can easily determine the object’s orientation angle, and human beings can recognize the object’s orientation angle more accessible. Accordingly, we can conclude that as the length of the object increases along a certain direction, the corresponding complex cell’s spike rate increases, and vice versa. This conclusion is consistent with the experimental phenomena observed by Hubel in rabbit cortical cells [11].

Examining the objects in Figure 16a, we can also find that when the object approaches to be a square, it is more and more difficult for human beings to identify the object’s orientation angle. Furthermore, from Figure 16b, we can observe that the activation of the $90^{\circ }$-selective complex cell tends to be equal to the $0^{\circ }$-selective complex cell. It is similar to our visual recognition mechanism. The closer the shape of an object is to a square, the more difficult it is to identify the orientation angle. This result also supports the previous conclusion in the last paragraph. In addition, the objects in the images were located at different locations, but the orientation angle could be detected correctly. This result shows the similarity with complex cells’ feature, which is responding selectively to stimuli with a particular orientation but no limit to the exact location of the stimulus [10,11].

To compare the performance of our AVS with CNN in orientation detection tasks and their noise resistance, we conducted a series of comparative experiments. First, we generated an original dataset, which consisted of 13,438 images (sized as 32 × 32), and the objects in each image had at least 32 pixel light spots. Each object had a specific orientation angle and location. Then based on the original dataset, we added noise and generated datasets with different types and quantities of noise. According to the arrangement of adding noise, they can be divided into two categories of noise data. Examples of two types of noise are shown in Figure 17. As shown in Figure 17a, the first type of noise was the case of no noise in the object, and the noise was randomly added to the background. The noise was randomly added to the whole image in the second type, as shown in Figure 18b. Then, we generated seven datasets with different quantities of noise for each type of noise: 5%, 10%, 15%, 20%, 25%, and 30% (for a 32 × 32 image, a certain proportion of pixels of the whole image were noise). The performance of AVS on these noise datasets and the recognition results are shown in Table 2. Results in Table 2 show that AVS has better noise resistance on background noise than whole-image noise.

Take the two objects shown in Figure 11a and Figure 12a as examples. We added noise to the images and detected them. The detection results on noise images are shown in Figure 18 and Figure 19. In Figure 18a, the noise was only randomly added to the background. Comparing the detection results with the non-noise one, we know that although the noise increased the spike rate of other complex cells, the $135^{\circ }$-selective complex cell was still most activated. In Figure 19a, the noise was randomly added to the whole image. The object’s shape had been changed. From Figure 19c, we know $0^{\circ }$-selective complex cell had the most spike rate, the same as the detection result of the object in Figure 12a. Comparing the spike records shown in Figure 19b with the spike records in Figure 12b, it is evident that the continuous noise in the image affected the complex cells’ spike rate. The continuous noise in the background activated the $0^{\circ }$-selective simple cell and $135^{\circ }$-selective simple cell in local receptive fields, and the continuous noise in the object inhibited the activations of simple cells. In short, though the noise would affect the spike rates of complex cells, the orientation angle of an object still could be recognized when the proportion of noise was lower than a certain degree. Additionally, the noise in an object had more effect on detection results than the noise in the background.

We also compared the generalization performance and noise immunity of AVS and CNN on orientation detection. The structure of the CNN we used in these experiments is shown in Figure 20. We chose Adam as the optimizer. Thirty 3 × 3 filters were used in the convolution layer, and 2 × 2 max-pooling was used in the pooling layer. The output size of the first affine layer was 100, and the last layer finally outputted four values. The training set consists of 10,750 ideal object images. We trained the CNN model 50 epochs and chose the model with the best detection accuracy as the final model. The ideal object testing dataset consists of 2687 images. We also collected eight natural objects (binary form). We rotated, moved the location, and changed the size of these objects within the image to obtain a natural object dataset that consists of 1280 different images. Figure 21 shows several examples of natural object data. Then based on the two original testing sets, we further generated several noise datasets.

The testing results are summarized in Table 3 and Table 4. From Table 3, we know that CNN’s recognition accuracy was very high without noise but dropped quickly with the proportion of noise increased. When the noise proportion exceeded 1%, the performance of CNN on ideal objects and natural objects all collapsed. For the AVS, it always kept an excellent advantage over CNN. Its recognition accuracy was still about 98% when tested on the ideal object datasets and could reach over 90% when tested on natural object datasets. Overall, the AVS can successfully give correct discrimination to objects’ orientation, regardless of the object’s shape, size, and location. Although the AVS already has an acceptable and good performance on natural objects, we further explored AVS’s performance and the impactors on AVS’s robustness. We recorded the classification results by AVS on 0% and 10% natural object noise datasets and plotted the corresponding confusion matrix, as shown in Figure 22. When the objects are without noise, AVS could give correct classifications to all objects. When objects are with noise, according to the confusion matrix, we can know that AVS still could provide accurate classifications to all $0^{\circ }$ and $90^{\circ }$ orientational objects, but had errors with some $45^{\circ }$ and $135^{\circ }$ orientational objects. Recalling the images with classification errors, we found that the objects are the same objects with different positions and sizes. These objects had activated a close number of $45^{\circ }$ and $135^{\circ }$ selective simple cells. When the images are clean, AVS can give the correct detection results, but when noise is added, the cells’ activation is affected, thus due to the classification error. Overall, AVS performs excellently on clean images and has good noise immunity on noise data.

## 4. Discussion

This research aimed to study the feasibility and reliability of the Hubel–Wiesel (HW) model, the concise and efficient quantitive methods of simple cells and complex cells, and realize an artificial visual system (AVS) based on the HW model for practicality. We realized the AVS with the following merit:Effectiveness; AVS could achieve 100% accuracy on ideal shape object datasets and natural objects with a particular orientation, which showed that our detection mechanism and the mechanism-based AVS could effectively detect the orientation of an object with distinct locations and sizes. The simulation results of biology-inspired experiments also showed that AVS is effective and highly consistent with real physiological experiments [12].Robustness; compared with CNN on orientation detection tasks, AVS costs fewer computation resources than CNN but has better performance and noise resistance.Interpretability; the mechanism, structure, and parameters of the AVS for global orientation detection were all designed from HW physiological model, so AVS does not need learning and saves many training resources. The CNN method is a black-box operation and usually requires more training data or deeper networks to improve noise immunity and has requirements on input data’s size. The AVS method does not need more layers and is easier to be accepted and trust. The calculation of the AVS is straightforward, and the image size is no need to fit the AVS so that its hardware implementation is also more straightforward than that of CNN. Even if we want to train AVS, we can use the perceptron algorithm instead of the MP neuron model. The AVS training can start from a better and reasonable initial condition to accelerate the learning process and prevent local minimums.

Overall, on the object-orientation detection tasks, AVS is much better than the CNN method because the AVS has good generalization ability, higher recognition accuracy, and stronger noise resistance and is explainable, feasible, reasonable, and robust. The AVS based on the HW model is feasible, efficient, and very similar to the perception mechanism of the biological visual system. Therefore, the implementation scheme of simple and complex cells realized in computer simulations is expected to provide a more helpful experiment direction in neural research.

## 5. Conclusions

This paper proposed a two-dimensional global orientation detective mechanism based on the Hubel–Wiesel (HW) model. Though we still know little about the principle of visual perception, we referred to the characteristics of the cortical cells with orientation selectivity. We designed four types of simple cells and complex cells. Using simple cells to extract every local orientation information, and activations of all simple cells are converged to the corresponding type of complex cells, we can get the global orientation according to the complex cell with the most activation. Simple cells and complex cells are realized on the McCulloch-Pitts neuron model. Based on this scheme, we proposed an artificial visual system (AVS) for global orientation detection and tested its performance on different orientation detection tasks. Although the inhibitory effect from the OFF region was omitted in simple cells, the success of AVS provided a possible scheme to explain the principles of orientation selectivity of cortical cells and gives evidence of the reasonability of the HW model, and also can provide a potential neural experiment implementation scheme on orientation selectivity research. Since the present AVS version can only detect those objects with a definite orientation and binary forms, future studies will need to extend the application and generalization on color images and more orientations.

## Figures and Tables

**Figure 1 brainsci-12-00470-f001:**
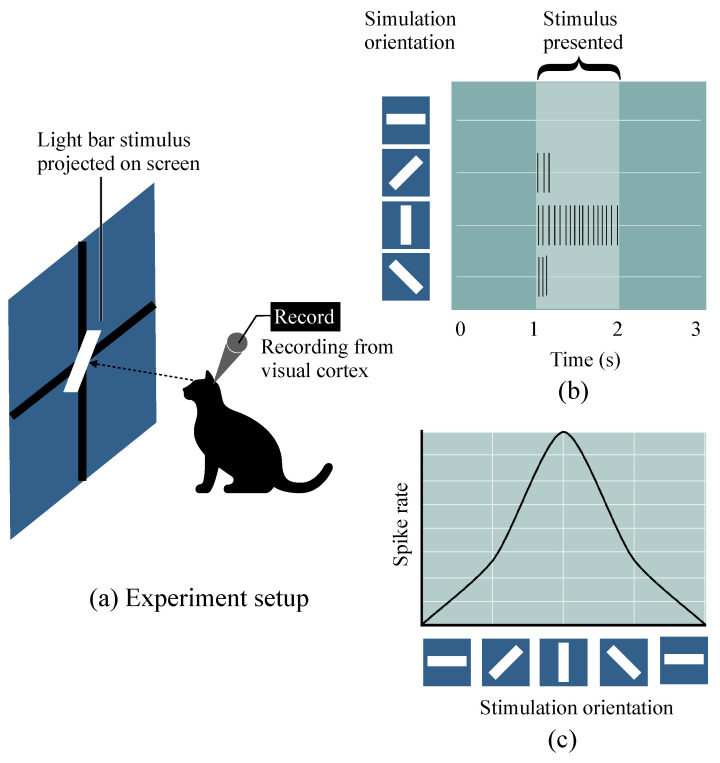
A neuron in the primary visual cortex responds selectively to line segments. (**a**) An anesthetized cat is fitted with contact lenses to focus the eyes on a screen, then project images on screen and record neuron responses by an extracellular electrode. (**b**) The neuron recorded in the primary visual cortex typically responds vigorously to a bar of light oriented at a particular orientation angle and with little or no response to other orientations. (**c**) The curve of neuron spike rate with the stimulation orientation changes.

**Figure 2 brainsci-12-00470-f002:**
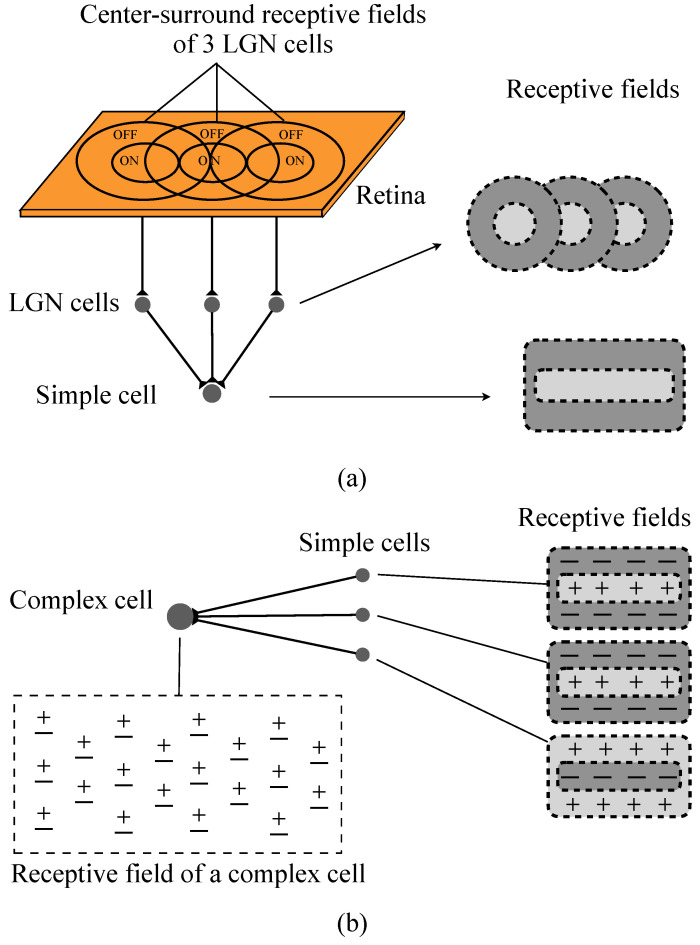
The convergent procession of simple cells’ and complex cells’ receptive fields. (**a**) A simple cell’s receptive field is formed by LGN cells’ spatially adjacent receptive fields. (**b**) Complex cell’s receptive field is an overlapping ON and OFF region which converged from simple cells’ receptive fields that with same orientation.“+”: ON region; “−”: OFF region.

**Figure 3 brainsci-12-00470-f003:**
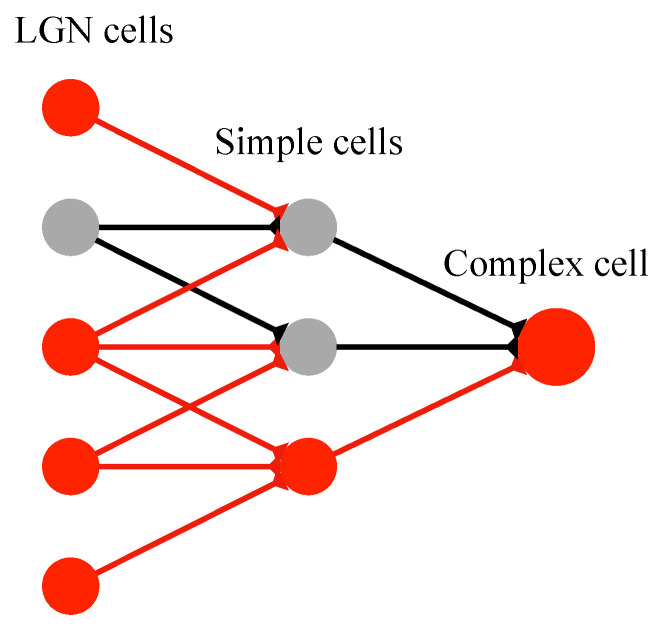
Hubel–Wiesel feedforward model. Effective synapse connections and activated cells are colored red.

**Figure 4 brainsci-12-00470-f004:**
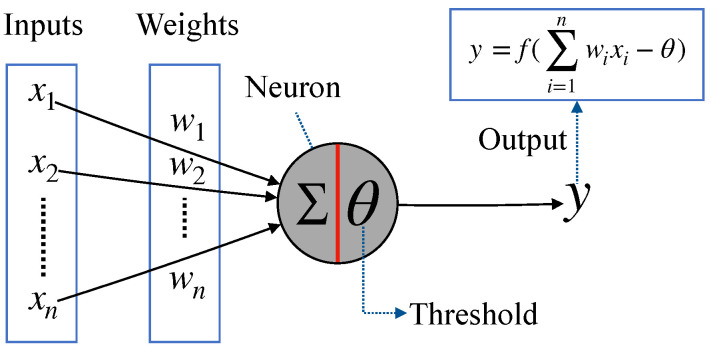
The structure of McCulloch-Pitts neuron model.

**Figure 5 brainsci-12-00470-f005:**
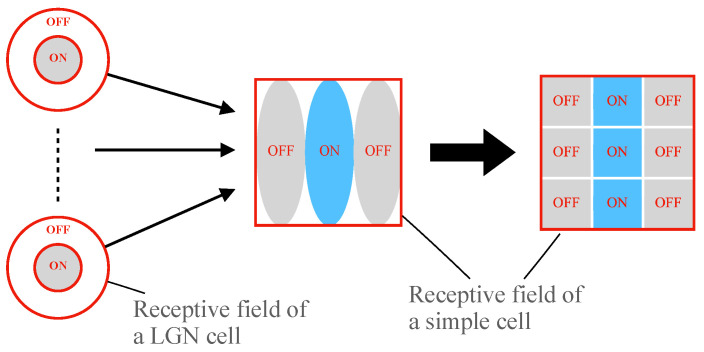
The formation of a simple cell receptive field by linking several LGN cell receptive fields.

**Figure 6 brainsci-12-00470-f006:**
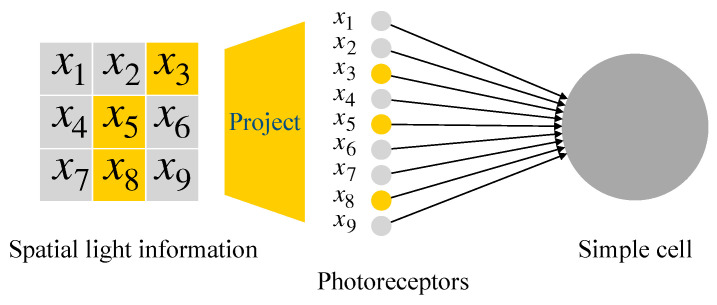
Signal transmission flow from light information to a simple cell. In a 3×3 region, the locations of each pixel are labeled from $x_{1}$ to $x_{9}$.

**Figure 7 brainsci-12-00470-f007:**
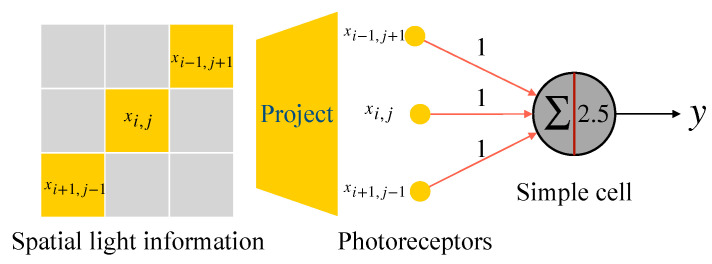
Realization of a $45^{\circ }$-selective simple cell based on MP model.

**Figure 8 brainsci-12-00470-f008:**
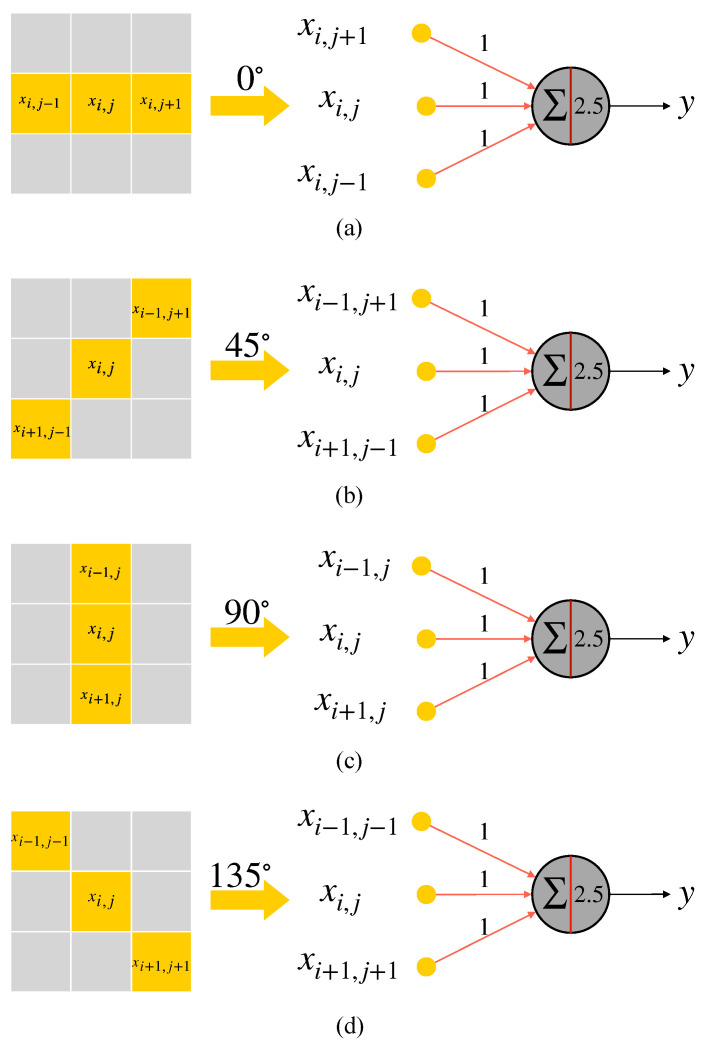
Four types of orientation-selective simple cells and their optimal stimuli orientation. (**a**) $0^{\circ }$-selective simple cell. (**b**) $45^{\circ }$-selective simple cell. (**c**) $90^{\circ }$-selective simple cell. (**d**) $135^{\circ }$-selective simple cell.

**Figure 9 brainsci-12-00470-f009:**
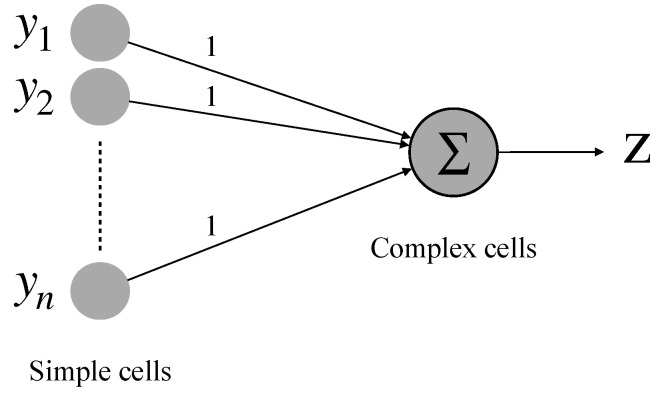
Realization of a complex cell.

**Figure 10 brainsci-12-00470-f010:**
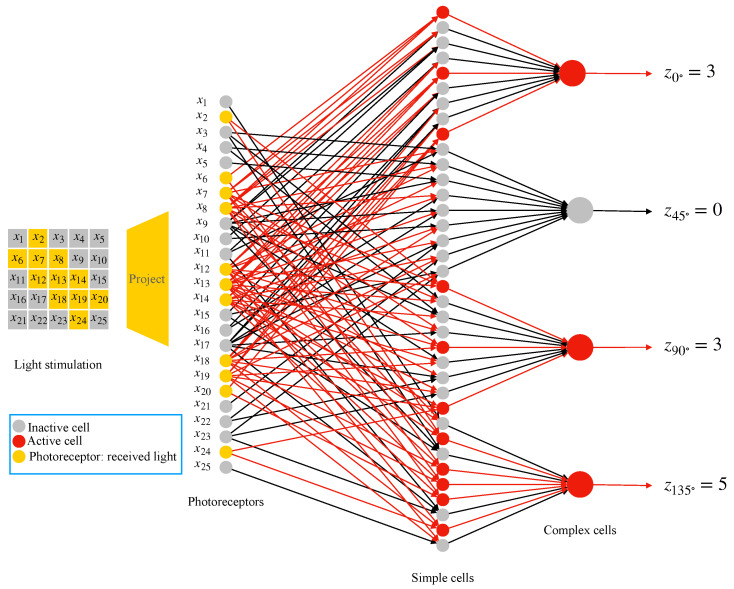
The structure of AVS used on detecting a 5×5 image. Effective neural connections and active cells are colored red. Photoreceptors received light are colored yellow.

**Figure 11 brainsci-12-00470-f011:**
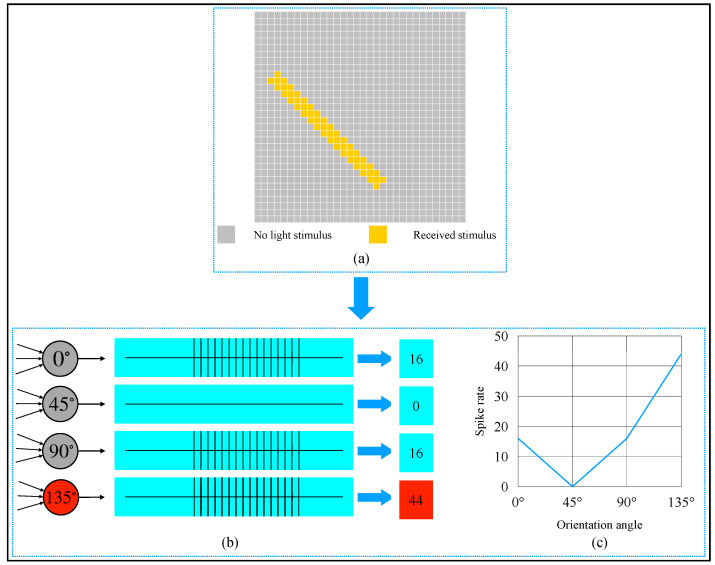
Computer simulation results of orientation detection on an object with a $135^{\circ }$ orientation angle. (**a**) Object. (**b**) Spike records. (**c**) Spike rate curve of four types of complex cells.

**Figure 12 brainsci-12-00470-f012:**
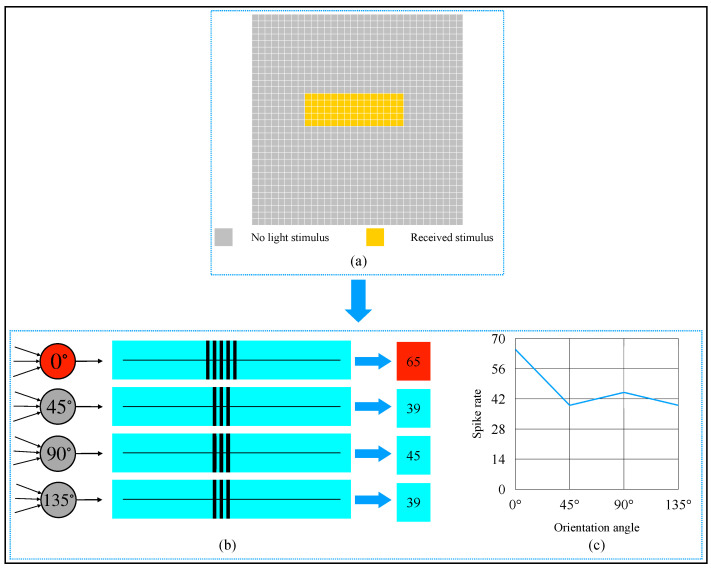
Computer simulation results of orientation detection on an object with $0^{\circ }$ orientation angle. (**a**) Object. (**b**) Spike records. (**c**) Spike rate curve of four types of complex cells.

**Figure 13 brainsci-12-00470-f013:**
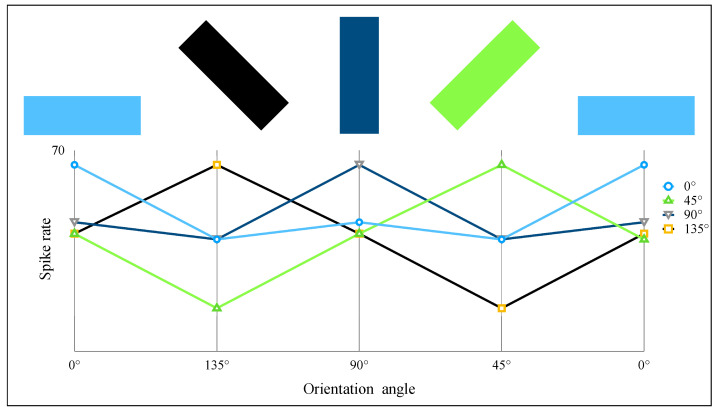
Spike rate of complex cells on the same size object when oriented toward different orientations ($0^{\circ },45^{\circ },90^{\circ }$, and $135^{\circ }$).

**Figure 14 brainsci-12-00470-f014:**
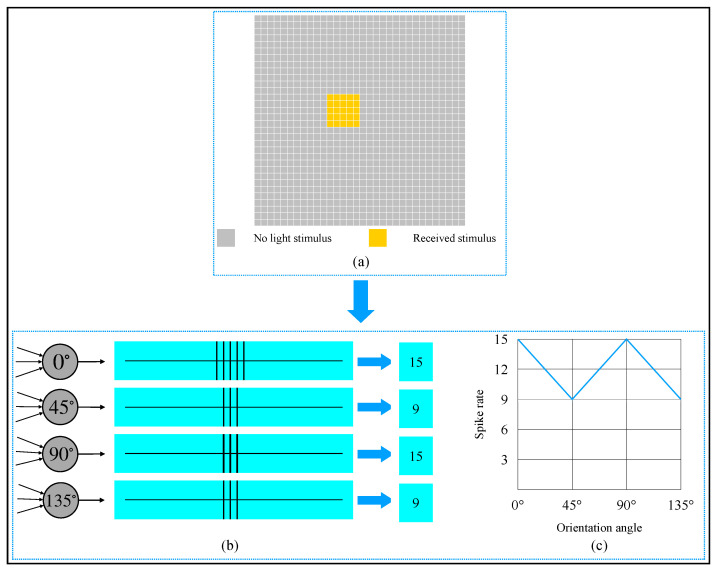
Computer simulation results of orientation angle detection on a square. (**a**) Object. (**b**) Spike records. (**c**) Spike rate curve of four types of complex cells.

**Figure 15 brainsci-12-00470-f015:**
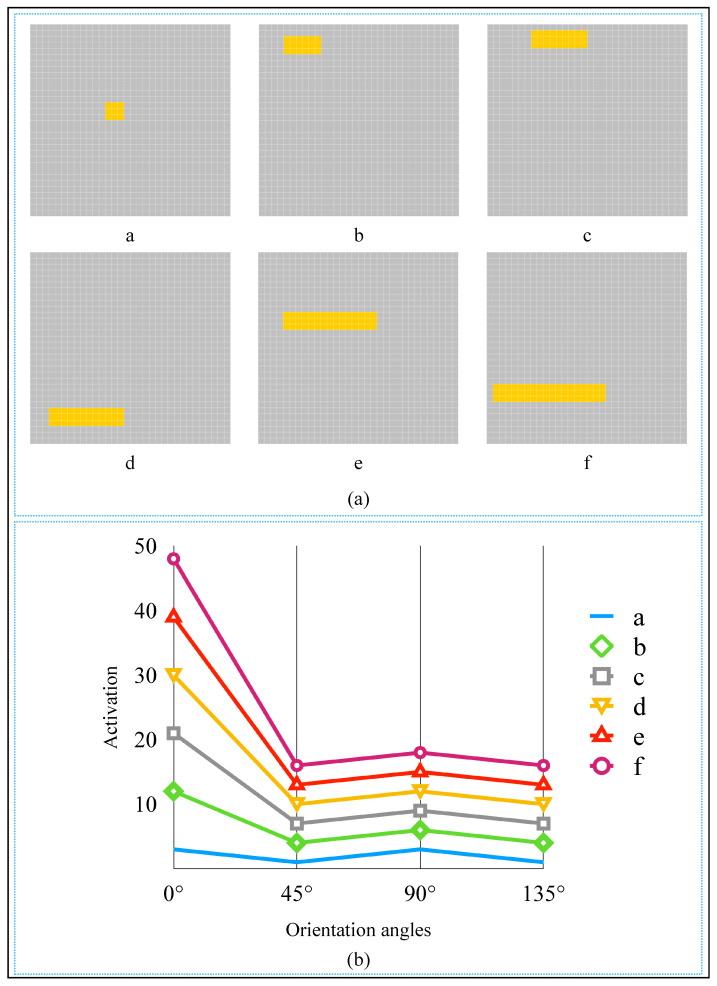
The detected objects and activation curves. (**a**) The objects to be detected. (**b**) Spike rate curves of complex cells on six objects.

**Figure 16 brainsci-12-00470-f016:**
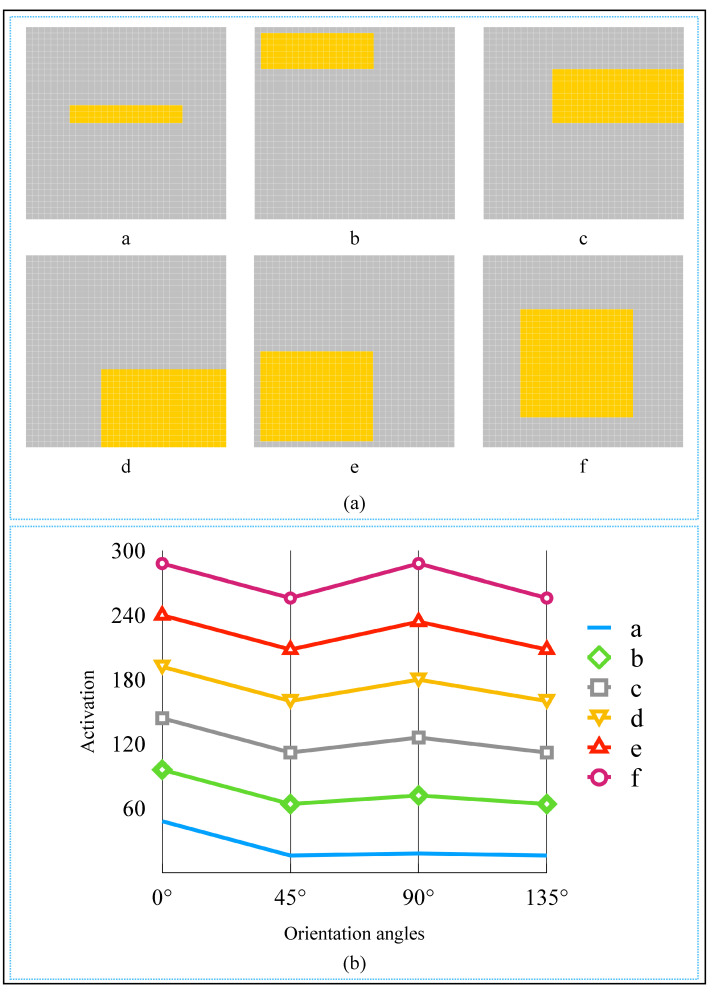
The detected objects and activation curves. (**a**) The objects to be detected. (**b**) Spike rate curves of complex cells on six objects.

**Figure 17 brainsci-12-00470-f017:**
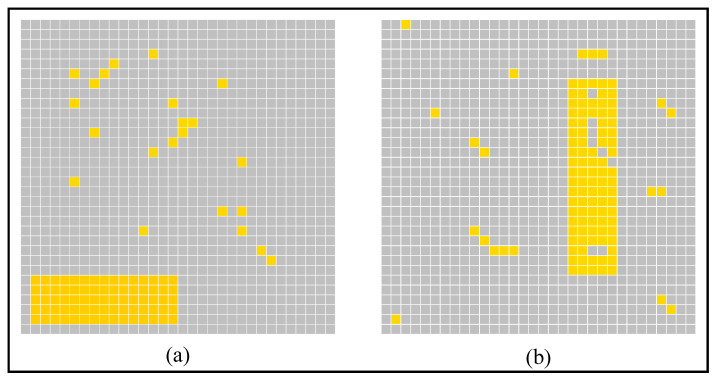
Two types of noise. (**a**) Random noise only in the background. (**b**) Random noise in the whole image.

**Figure 18 brainsci-12-00470-f018:**
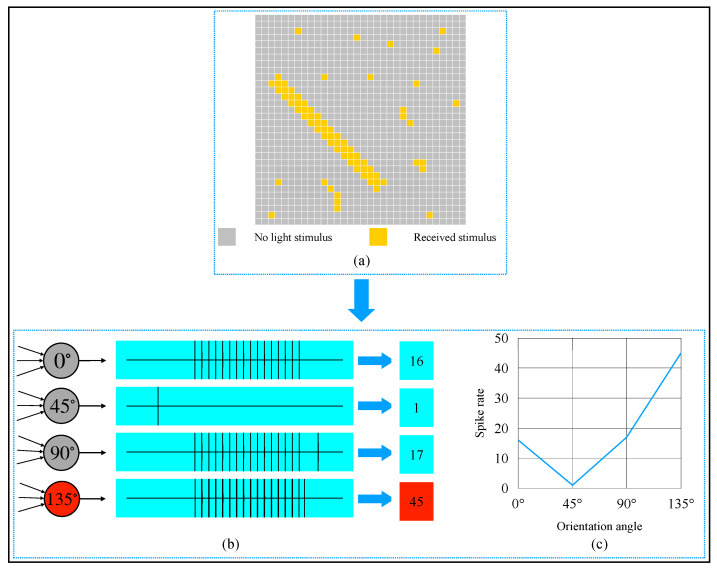
Computer simulation results of orientation detection on an object with $135^{\circ }$ orientation angle. (**a**) Object and image noise. (**b**) Spike records. (**c**) Spike rate curve of four types of complex cells.

**Figure 19 brainsci-12-00470-f019:**
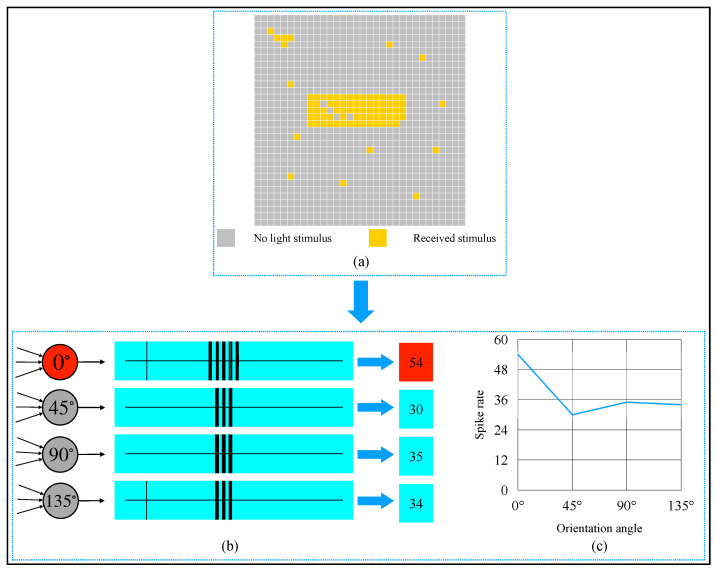
Computer simulation results of orientation detection on an object with $0^{\circ }$ orientation angle. (**a**) Object and image noise (**b**) Spike records. (**c**) Spike rate curve of four types of complex cells.

**Figure 20 brainsci-12-00470-f020:**
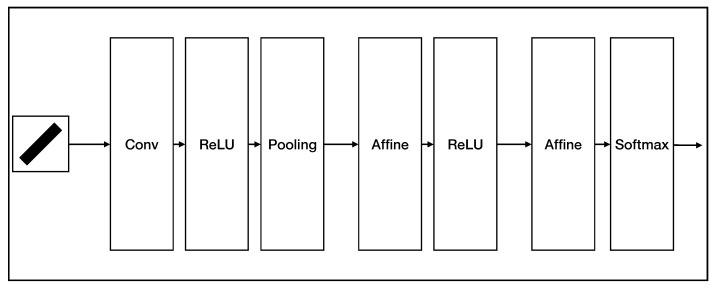
The structure of CNN used for orientation detection.

**Figure 21 brainsci-12-00470-f021:**
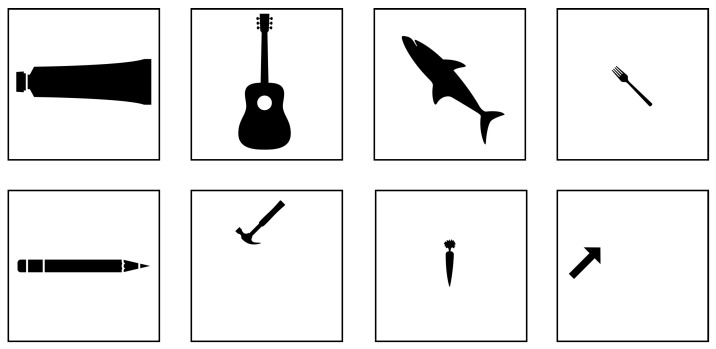
Natural objects.

**Figure 22 brainsci-12-00470-f022:**
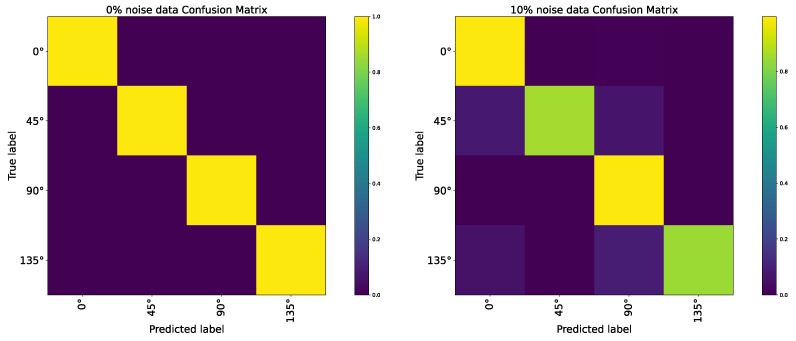
Confusion matrix.

**Table 1 brainsci-12-00470-t001:** Detection accuracy analysis of the AVS on binary image dataset.

Image Data		0${}^{\circ }$	45${}^{\circ }$	90${}^{\circ }$	135${}^{\circ }$
	No. of samples	960	900	960	900
3 pixel	Correct numbers	960	900	960	900
	Accuracy	100%	100%	100%	100%
	No. of samples	928	841	928	841
4 pixel	Correct numbers	928	841	928	841
	Accuracy	100%	100%	100%	100%
	No. of samples	1699	2249	1699	2249
8 pixel	Correct numbers	1699	2249	1699	2249
	Accuracy	100%	100%	100%	100%
	No. of samples	2379	3411	2379	3411
12 pixel	Correct numbers	2379	3411	2379	3411
	Accuracy	100%	100%	100%	100%
	No. of samples	1319	1489	1319	1489
16 pixel	Correct numbers	1319	1489	1319	1489
	Accuracy	100%	100%	100%	100%
	No. of samples	1284	1645	1284	1645
32 pixel	Correct numbers	1284	1645	1284	1645
	Accuracy	100%	100%	100%	100%
	No. of samples	2515	1275	2515	1275
≥48 pixel	Correct numbers	2515	1275	2515	1275
	Accuracy	100%	100%	100%	100%

**Table 2 brainsci-12-00470-t002:** The recognition accuracy of AVS under two noise conditions.

Type of Noise	Proportion of Noise						
	0%	5%	10%	15%	20%	25%	30%
Background noise	100%	100%	100%	99.911%	98.571%	95.289%	91.851%
Whole-image noise	100%	99.970%	98.772%	95.036%	87.602%	78.382%	67.771%

**Table 3 brainsci-12-00470-t003:** The recognition accuracy of AVS and CNN on ideal object datasets.

Method	Proportion of Noise										
	0%	1%	2%	3%	4%	5%	6%	7%	8%	9%	10%
AVS	100%	100%	100%	100%	99.963%	99.963%	99.926%	99.739%	99.330%	99.032%	98.772%
CNN	100%	85.225%	55.862%	43.431%	39.747%	37.514%	36.323%	34.090%	32.006%	29.996%	29.028%

**Table 4 brainsci-12-00470-t004:** The recognition accuracy of AVS and CNN on natural object datasets.

Method	Proportion of Noise										
	0%	1%	2%	3%	4%	5%	6%	7%	8%	9%	10%
AVS	100%	99.609%	99.219%	98.906%	97.578%	97.109%	96.094%	95.547%	94.766%	92.813%	92.422%
CNN	92.813%	54.766%	48.705%	42.500%	38.281%	36.484%	34.922%	33.828%	32.734%	31.719%	31.406%

## Data Availability

The data presented in this study are available on request from the corresponding author. The data are not publicly available due to data privacy regulations.

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
