# Peer review of "Artificial Visual System for Orientation Detection Based on Hubel–Wiesel Model"

_brainsci, 2022, doi:10.3390/brainsci12040470_

Round 1

Reviewer 1 Report

This is interesting work; however, the author claim that the orientation sensitivity has not reported previously looks bit wrong. since there is a paper like, https://doi.org/10.1002/adma.201903095.
The author need to focus on the litratures more ciriticially. 

Author Response

Thank you very much for your comments and suggestion. We have modified our description of previous related work and cited the paper reviewer provide. 

‘However, the HW model still has not been fully proved physiologically, and there are few concise but efficient systems to quantify and simulate the HW model and can be used for object orientation detection applications.’

Reviewer 2 Report

I think the topic of this paper is very interesting, and the introduction of network model and structure is very detailed.

Deficiencies and suggestions are as follows:

1) The model given in the experiment is very simple. The shortcomings and the advanced methods of the proof can also solve similar problems in the same direction. It is suggested that the cases of complex scenes should be complements, including noise effects.

2) For the evaluation of recognition results, a simple accuracy comparison is adopted, and the accuracy index used in the field of machine vision should be supplemented.

Author Response

1. The model given in the experiment is very simple. The shortcomings and the advanced methods of the proof can also solve similar problems in the same direction. It is suggested that the cases of complex scenes should be complements, including noise effects.

Response: Thank you very much for your comments and suggestion. For the cases of complex scenes, we collect eight natural objects (binary form) and rotate, move the location, and change the size of these objects within the image to obtain a natural object dataset that consists of 1,280 different images. Then based on the original set, we further generated several noise datasets. We validated the noise immunity of AVS and trained CNN on these natural object datasets. In this paper, our AVS is a binary version. In future research, we can extend the AVS to a grayscale version for application in more complex scenes.

2. For the evaluation of recognition results, a simple accuracy comparison is adopted, and the accuracy index used in the field of machine vision should be supplemented.

Response: Thank you very much for your comments and suggestion. We further introduced the confusion matrix method to evaluate AVS’s performance on natural object datasets and the impactors on classification errors. These contents are discussed in the last paragraph of section 3.

Reviewer 3 Report

I have the following recommendations regarding improvements of the paper.

1. Add the motivation of the proposed approach.
2. Literature related to the proposed scheme published in the last 3 years should be added (at least 3 more references).
3. Write the Research gap i.e. which mentions the limitations in the literature and how the authors overcome it.
4. Problem statements need to be written that clearly illustrate which problem is the focus of the study.
5. Research contributions, which clearly describe the novelty of the proposed solution should be added in the form of bullets or numbered form (1, 2, 3, ...).
6. Mention some future directions and limitations of the proposed scheme in the conclusions section.
7. Mention all the acronyms in a table by the end of the manuscript.
8. Paper should be checked for english spellings/minor grammatical mistakes.
9. Improve the figures with different fill/colors to clearly distinguish one scheme from the others.

Author Response

1. Add the motivation of the proposed approach.

Response: Thank you very much for your suggestion. We have emphasized our research motivations in the Introduction section.

2. Literature related to the proposed scheme published in the last 3 years should be added (at least 3 more references).

Response: Thank you very much for your suggestion. We have added several related papers published in the last 3 years.

3. Write the Research gap i.e. which mentions the limitations in the literature and how the authors overcome it.

Response: Thank you very much for your suggestion. We have discussed the related works in the Introduction section.

4. Problem statements need to be written that clearly illustrate which problem is the focus of the study.

Response: Thank you very much for your comments and suggestion. We aim to propose a concise but efficient method to simulate the HW model and use it for application. We have emphasized the focus of our study in the Introduction section. ‘So far, for the classical HW model, we lack a concise and efficient quantitive manner.’ ; ‘To prove the reasonability and practicality of the HW model and explain the orientation selectivity in a quantitive manner, we propose a McCulloch-Pitts (MP) neuron-based orientation detective scheme and implement an artificial visual system (AVS) for two-dimensional object orientation detection. ’

5. Research contributions, which clearly describe the novelty of the proposed solution should be added in the form of bullets or numbered form (1, 2, 3, …).

Response: Thank you very much for your comments and suggestion. We have discussed the AVS with bullet list form in the Discussion section.

6. Mention some future directions and limitations of the proposed scheme in the conclusions section.

Response: Thank you very much for your comments and suggestion. We have pointed out the future directions and limitations of the proposed scheme in the Conclusions section.

7. Mention all the acronyms in a table by the end of the manuscript.

Response: Thank you very much for your comments and suggestion. We have mentioned the abbreviation we used at the end of the manuscript.

8. Paper should be checked for english spellings/minor grammatical mistakes.

Response: Thank you very much for your comments and suggestion. We have improved the English and corrected the typo.

9. Improve the figures with different fill/colors to clearly distinguish one scheme from the others.

Response: Thank you very much for your comments and suggestion. We have improved the display of Figures 13, 15, and 16.
